# Primary Orbital Extranodal Natural Killer/T-Cell Lymphoma, Nasal Type, without Nasal Involvement

**DOI:** 10.3390/jcm11237010

**Published:** 2022-11-27

**Authors:** Dalan Jing, Debo You, Ziyuan Liu, Wei Wang

**Affiliations:** 1Department of Ophthalmology, Peking University Third Hospital, Beijing 100191, China; 2Beijing Key Laboratory of Restoration of Damaged Ocular Nerve, Peking University Third Hospital, Beijing 100191, China

**Keywords:** extranodal natural killer/T-cell lymphoma, histopathological examination, immunohistochemical, epstein-barr virus-encoded small RNA

## Abstract

Extranodal natural killer/T-cell lymphoma (ENKL) usually occurs in the nose or the nasopharynx, but extranasal and disseminated disease can occur. In this disease, orbital involvement is more commonly seen, but without nasal involvement is rare. A 61-year-old woman was referred with a one-month history of a remarkably enlarging salmon-colored mass arising in the right inner canthus, with redness and painlessness. The motility of the right eye was limited in the medial direction, with external deviation of the eyeball. A magnetic resonance imaging (MRI) scan of the orbits showed a mass of irregular shape located in the right inner canthus, without any sinus involvement. A histopathological examination concluded a diagnosis of primary orbital extranodal natural killer/T-cell lymphoma, nasal type. Her orbital mass significantly reduced to near disappeared after chemotherapy. From the first visit to the present, the survival duration of this patient was more than 1 year. This patient was still alive with a high quality of life and with no systemic metastasis. Extranodal natural killer/T-cell lymphoma, nasal type may primarily arise in the orbit without nasal involvement. Early discovery, early biopsy and diagnosis and early appropriate treatment can successfully control tumors and improve prognosis.

## 1. Introduction

Non-Hodgkin lymphoma (NHL) is the most common malignant tumor involving the orbit. Within this group, natural killer T-cell lymphoma (NKTCL), a rare form of NHL, is an aggressive disease with a high rate of mortality [1]. Extranodal natural killer/T-cell lymphoma (ENKTL) is a type of NKTCL that usually occurs in the nose or the nasopharynx, but extranasal and disseminated disease can occur [2,3], while a few cases of primary orbital or intraocular ENKTL have been reported [4,5]. In nasal NKTCL, orbital involvement is more commonly seen, but without nasal involvement is rare. Pathologic examination rather than affected sites can be helpful in making the diagnosis. We report a rare case of ENKL that presented as the primary orbital involvement without any systemic involvement.

## 2. Case Presentation

In November 2016, a 61-year-old woman was referred to our hospital with a one-month history of a remarkably enlarging mass arising in the right inner canthus, with redness and painlessness. She was treated with antibiotic eye drops (specific medication is unknown) in a local community hospital, resulting in no significant improvement. The patient was previously healthy with no history of familial hereditary diseases.

On examination, the corrected visual acuity was 20/30 in right eye and 20/20 in left eye. The intraocular pressure (IOP) was 14 and 12 mmHg, respectively. No afferent pupillary defect was noted in either eye. The motility of the right eye was limited in the medial direction, with external deviation of the eyeball. A small amount of purulent discharge was noted in the right eye, and marked chemosis was noted medially. There was an obvious elevated salmon-colored mass in the right inner canthus, which protruded to the palpebral fissure and was accompanied by superficial ulceration, and its border was ill-defined. The superomedial and inferior aspects of the orbit became involved (Figure 1). Anterior and posterior segment examination was unremarkable. The palpation of superficial lymph nodes, liver and kidney had no obvious abnormality.

Computed tomography scan imaging (CT) of the orbits revealed space-occupying lesions in the right medial orbit and no sign of nasal infiltration (Figure 2). Only after 6 days, a magnetic resonance imaging (MRI) scan of the orbits showed a mass of irregular shape that appeared isointense to muscle on TI-weighted sequences, with mild hyperintensity to muscle on T2-weighted sequences. There is marked homogeneous enhancement in the mass. It was located in the right inner canthus, which enveloped the eyeball from the 1–7 o’clock position, the medial rectus and the inferior rectus, without any nasal involvement (Figure 3). Compared to CT, the mass significantly increased. With the presumptive diagnosis of lymphoma, the ocular mass was partly resected, and frozen section examination during surgery suggested lymphoproliferative disease and potential lymphoma.

Histopathological examination of the specimen revealed a diffuse lymphoid infiltration composed of atypical lymphocytes with hyperchromic nuclei (Figure 4A). Immunohistochemical studies were positive for antibodies against cytoplasmic CD3 (Figure 4B), CD56 (Figure 4C) and T-cell-restricted intracellular antigen (TIA-1) (Figure 4D) but negative for CD20, CyclinD1, PAX-5, CD10, CD30, CD15, MUM1, Bcl-6, CD138, Kappa, CD23, and CD163. Some of the neoplastic cells showed expression of CD21, CD5 and Bcl-2. The proliferation index by Ki-67 labeling was more than 80%. In situ hybridization for Epstein-Barr virus (EBV)-encoded small RNA (EBER) was positive, indicated the integrity of EBV mRNA in the tissue samples. On the basis of these findings, primary orbital extranodal natural killer/T-cell lymphoma nasal type was diagnosed. In addition, the EBV DNA copy number was less than 500 copies/mL. Disease is progressing rapidly.

The patient was referred to the Department of Hematology for special treatment. The B-scan of whole-body superficial lymph nodes indicated no masses. An 18F-fluorodeoxyglucose (18F-FDG) PET scan showed increased uptake of space-occupying lesions in the right eye but failed to show any systemic involvement. Furthermore, the bone marrow biopsy was roughly normal.

In the department of hematology, the patient received 2 cycles of GEMOX (gemcitabine, oxaliplatin, dexamethasone) +Pegaspargase chemotherapy, resulting in improvement of ocular motility and correction of external deviation. However, after these treatments, she had steroid diabetes. A PET scan demonstrated that the previous orbital mass significantly reduced and nearly disappeared. In addition, the corrected visual acuity of right eye returned to 20/20, and the eyeball could move freely in all directions. The patient continued to receive another 3 cycles of GEMOX + pegaspargase and 1 cycle of CHOP-E (cyclophosphamide, epirubicin, vindesine, etoposide, prednisone acetate) + pegaspargase chemotherapy. After treatment, the patient was in complete remission (CR) according to a PET scan. After chemotherapy, 60 Gy radiotherapy was given sequentially in 30 fractions to the right orbit. Until November 2021, this patient was still alive with a higher quality of life and with neither local recurrence nor systemic metastasis.

## 3. Discussion and Conclusions

NHLs are the most common malignant tumors comprising the orbit. Compared with mucosa-associated lymphoid tissue (MALT) lymphoma, NKTCL has completely different characteristics, with low incidence, rapid progression, and a high fatality rate. Extranodal NKTCL is a highly aggressive lymphoproliferative uncommon disorder associated with Epstein-Barr virus infection, which typically arises from the nasal cavity and paranasal sinuses (70%), other affected areas involving the skin, salivary glands, liver, lymph nodes, gastrointestinal tract, lungs, and the testis [3,6,7,8,9,10]. Extranodal NKTCL is known for poor survival prognosis, with a mean survival of 12.5 months; once involved in the orbit, the median survival is only 4 months [6,9,11]. Extranodal NKTCL is relatively easy to consider when the lesions are simultaneously found in both the nasal cavity and adjacent orbit. When only the orbit is involved, it is usually misdiagnosed. Primary ocular extranodal NKTCL is quite rare and limited to a few case reports. Coupland first reported 2 cases of primary orbital NKTCL without nasal mucosal involvement [11]. Since then, 18 primary ocular extranodal NKTCL cases have been reported [12,13]. Conjunctival chemosis, proptosis, limited ocular movement, etc., are the initial symptoms of orbital NKTCL. All of these manifestations may be misdiagnosed as orbital pseudotumor or orbital cellulitis, delaying the early correct treatments [6,11]. However, compared to orbital pseudotumor and orbital cellulitis, the features of orbital NKTCL are salmon-colored lesions, rapid progression, no infective symptoms, negative blood tests, and no efficacy of antibiotics or steroids. Despite these findings, the clinical diagnosis of orbital NKTCL is still difficult.

Imaging of the orbit by CT or MRI scanning is recommended when the diagnosis is uncertain. The invasion of nasal passages and maxillary sinuses are common signs of nasal NKTCL. However, compared with other lymphoproliferative diseases, the imaging features of nasal NKTCL are nonspecific. CT and MR imaging are supplementary examinations to diagnose and evaluate the stage of nasal NKTCL [14]. In addition, PET scans play an important role in assessing systemic metastasis and the prognosis of the lesion.

Early biopsy is the most important procedure when the diagnosis of lymphoma is considered. It plays a crucial role in definitive diagnosis and prognosis. Through our case and literature reading, the essentials of biopsy include the removal of more tissues near the center of the lesion and one more biopsy to highly suspected cases whose first biopsy was negative. The typical histologic characteristics of nasal NKTCL are angioinvasion, angio-destruction and angiocentric growth; prominent necrosis and frequent apoptosis; and blastic appearance of the neoplastic cells [9,15]. Immunohistochemical analysis was positive for CD2, CD16, CD56, CD57, cytoplasmic CD3ε, and cytotoxic molecules, such as granzyme B, TIA-1, and perforin. In situ hybridization for EBER is positive in all tumor cells [6,7]. In our case, the time from onset to biopsy was only 1 month, which may be the main reason for the better prognosis compared with 5 weeks to 6 months in other studies. Early intervention may have been effective in this case.

In summary, primary orbital extranodal NKTCL without nasal involvement is a rare but serious illness that always has a poor prognosis. Early clinical manifestations are nonspecific, giving rise to delayed early diagnosis. From this case, when the lesion was suspected to be lymphoproliferative disease with characteristics of rapid progression, no infective symptoms, and no effect of antibiotics or steroid therapy, special types of lymphoma (MALT) or extramedullary leukemia should be considered. It is immediately apparent that orbital incisional biopsy is crucial to suspicious and refractory cases. Once the diagnosis of extranodal NKTCL is made, early chemotherapy or chemoradiotherapy can improve the prognosis and survival rates.

## Figures and Tables

**Figure 1 jcm-11-07010-f001:**
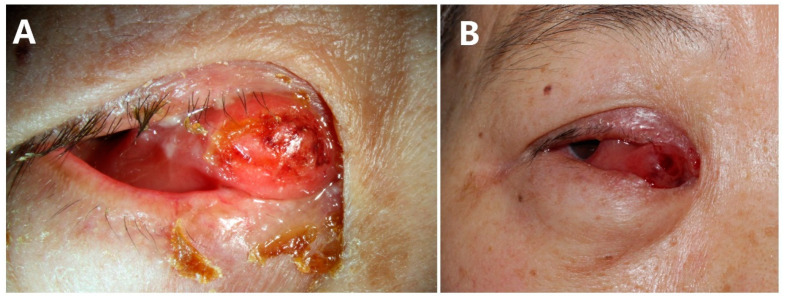
Extraocular manifestations. (**A**) Clinical features during the first visit to our hospital. Salmon-colored mass in the right inner canthus and external deviation of the eyeball. This was accompanied by a small amount of purulent discharge and superficial ulceration. (**B**) Before the operation and removal of the surface discharge.

**Figure 2 jcm-11-07010-f002:**
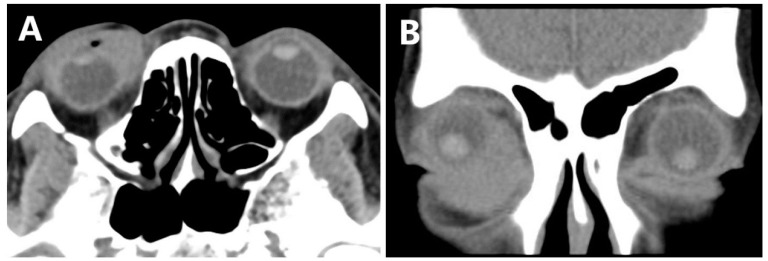
CT scan of the orbits. The mass was located in the right medial orbital, and there was no sign of nasal infiltration. Equal to soft tissue density. (**A**) Horizontal view. (**B**) Coronal view.

**Figure 3 jcm-11-07010-f003:**
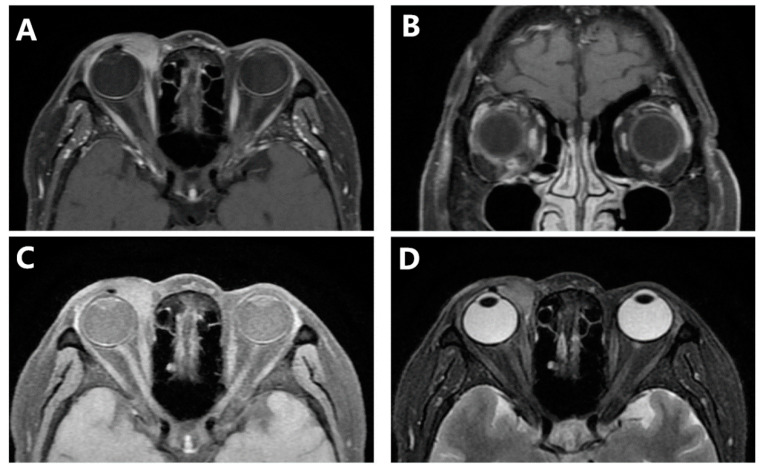
An MRI scan of the orbits showed a mass of irregular shape that appeared isointense on TI-weighted sequences located in the right medial orbital. (**A**) Horizontal view of contrast-enhanced TI-weighted fat-depressed sequences. (**B**) Coronal view on contrast-enhanced TI-weighted sequences. (**C**) Horizontal view of TI-weighted fat-depressed sequences. (**D**) Horizontal view of T2-weighted fat-depressed sequences.

**Figure 4 jcm-11-07010-f004:**
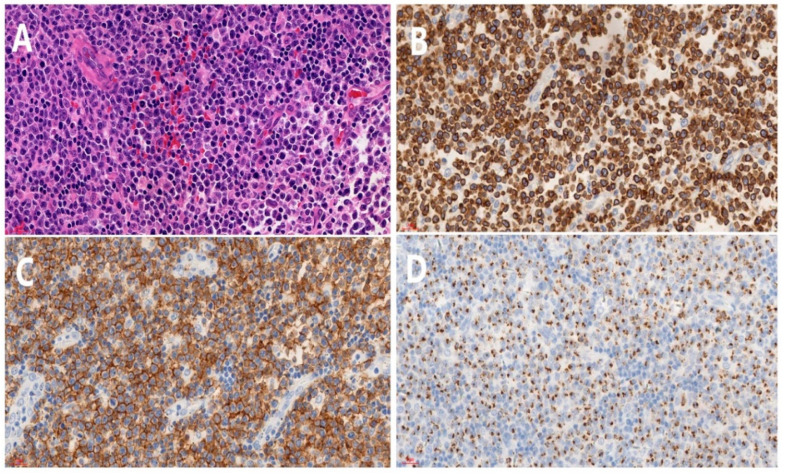
Histopathological examination of specimen (**A**) shows diffuse atypical lymphocyte infiltration (H&E) (original magnification ×40). (**B**) shows staining for CD3 (original magnification ×40). (**C**) Staining for CD56 (original magnification ×40). (**D**) Staining for TIA-1 (original magnification ×40).

## Data Availability

All the data supporting our findings are contained within the manuscript.

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
