# Peer review of "Primary Orbital Extranodal Natural Killer/T-Cell Lymphoma, Nasal Type, without Nasal Involvement"

_jcm, 2022, doi:10.3390/jcm11237010_

Round 1

Reviewer 1 Report

The report is well presented with good clinical detail, clinical photos and microscopic photos. The discussion is longer than necessary, including too many details about prior case reports.

Minor corrections to the text are needed throughout. Examples: line 88 uses wrong verb tense, line 107 repeats "after chemotherapy", line 122 uses "tests" instead of "testis", line 156 uses "plastic" instead of "blastic".

Author Response

Dear Reviewers:

Thank you very much for your letter and for the reviewers’ comments concerning our manuscript entitled “Primary Orbital Extranodal Natural Killer/T-Cell Lymphoma, Nasal Type, Without Nasal Involvement” (Manuscript ID:jcm-2002946). We have revised manuscript in accordance with requirements of the reviewers. We have addressed the comments raised by reviewers, and the amendments are highlighted in red in revised manuscript. Those comments are all valuable and very helpful for revising.

Independent Review Report, Reviewer 1
1. Response to comment: The report is well presented with good clinical detail, clinical photos and microscopic photos. The discussion is longer than necessary, including too many details about prior case reports.

Response: Thank you for the comments.Your comments are important for us to improve the quality of articles.We read the discussion part and delete the lengthy part.

  1. Response to comment:Minor corrections to the text are needed throughout. Examples: line 88 uses wrong verb tense, line 107 repeats "after chemotherapy", line 122 uses "tests" instead of "testis", line 156 uses "plastic" instead of "blastic".

Response: Thank you for the comments.We have corrected the errors you pointed out.

Other changes: Since this magazine can only have two co first authors, but Dr. Liu Ziyuan has given support in providing fund support, preliminary construction and revision of articles, we list him as the co correspondent author.

We tried our best to improve the manuscript and made some changes in the manuscript. We appreciate for your warm work earnestly, and hope that the correction will meet with approval.

Once again, thank you very much for your comments and suggestions.

Reviewer 2 Report

This article showed extranodal NKT cell lymphoma without nasal involvement that was detected early by biopsy and treated successfully by chemotherapy and radiation, resulting long term survival. I agree that Extranodal Natural Killer/T-Cell Lymphoma, Nasal Type, Without Nasal Involvement is rare and difficult for diagnosis. However there still several articles that addressed NKT cell lymphoma without nasal involvement that the authors did not cite such as;

Maruyama K, Kunikata H, Sugita S, Mochizuki M, Ichinohasama R, Nakazawa T. First case of primary intraocular natural killer t-cell lymphoma. BMC Ophthalmol. 2015 Nov 19;15:169. doi: 10.1186/s12886-015-0158-0. PMID: 26585973; PMCID: PMC4653874.

Okada A, Harada Y, Inoue T, Okikawa Y, Ichinohe T, Kiuchi Y. A case of primary extranodal natural killer/T-cell lymphoma in the orbit and intraocular tissues with cerebrospinal fluid involvement. Am J Ophthalmol Case Rep. 2018 May 17;11:37-40. doi: 10.1016/j.ajoc.2018.05.002. PMID: 29978138; PMCID: PMC6026719.

According to these previous article, the authors need to show some novel findings from this case.

Author Response

Dear Reviewers:

Thank you very much for your letter and for the reviewers’ comments concerning our manuscript entitled “Primary Orbital Extranodal Natural Killer/T-Cell Lymphoma, Nasal Type, Without Nasal Involvement” (Manuscript ID:jcm-2002946). We have revised manuscript in accordance with requirements of the reviewers. We have addressed the comments raised by reviewers, and the amendments are highlighted in red in revised manuscript. Those comments are all valuable and very helpful for revising.

Independent Review Report, Reviewer 2

  1. Response to comment: This article showed extranodal NKT cell lymphoma without nasal involvement that was detected early by biopsy and treated successfully by chemotherapy and radiation, resulting long term survival. I agree that Extranodal Natural Killer/T-Cell Lymphoma, Nasal Type, Without Nasal Involvement is rare and difficult for diagnosis. However there still several articles that addressed NKT cell lymphoma without nasal involvement that the authors did not cite such as;

Maruyama K, Kunikata H, Sugita S, Mochizuki M, Ichinohasama R, Nakazawa T. First case of primary intraocular natural killer t-cell lymphoma. BMC Ophthalmol. 2015 Nov 19;15:169. doi: 10.1186/s12886-015-0158-0. PMID: 26585973; PMCID: PMC4653874.

Okada A, Harada Y, Inoue T, Okikawa Y, Ichinohe T, Kiuchi Y. A case of primary extranodal natural killer/T-cell lymphoma in the orbit and intraocular tissues with cerebrospinal fluid involvement. Am J Ophthalmol Case Rep. 2018 May 17;11:37-40. doi: 10.1016/j.ajoc.2018.05.002. PMID: 29978138; PMCID: PMC6026719.

According to these previous article, the authors need to show some novel findings from this case.

Response: We feel great thanks for your professional review work on our article. We carefully read these documents, included them in the references(reference 4,5), and made corresponding modifications.

Other changes: Since this magazine can only have two co first authors, but Dr. Liu Ziyuan has given support in providing fund support, preliminary construction and revision of articles, we list him as the co correspondent author.

We tried our best to improve the manuscript and made some changes in the manuscript. We appreciate for your warm work earnestly, and hope that the correction will meet with approval.

Once again, thank you very much for your comments and suggestions.

Round 2

Reviewer 2 Report

The authors addressed my comments.